# Proteomic Analysis of the Resistance Mechanisms in Sugarcane during *Sporisorium scitamineum* Infection

**DOI:** 10.3390/ijms20030569

**Published:** 2019-01-29

**Authors:** Pratiksha Singh, Qi-Qi Song, Rajesh Kumar Singh, Hai-Bi Li, Manoj Kumar Solanki, Mukesh Kumar Malviya, Krishan Kumar Verma, Li-Tao Yang, Yang-Rui Li

**Affiliations:** 1Agricultural College, State Key Laboratory of Subtropical Bioresources Conservation and Utilization, Guangxi University, Nanning 530005, China; singh.pratiksha23@gmail.com (P.S.); alice771992@126.com (Q.-Q.S.); lihaibi@gxaas.net (H.-B.L.); 2Guangxi Key Laboratory of Sugarcane Genetic Improvement, Key Laboratory of Sugarcane Biotechnology and Genetic Improvement (Guangxi), Ministry of Agriculture, Sugarcane Research Center, Chinese Academy of Agricultural Sciences; Sugarcane Research Institute, Guangxi Academy of Agricultural Sciences, Nanning 530007, China; rajeshsingh999@gmail.com (R.K.S.); mukeshmicro@rediffmail.com (M.K.M.); drvermakishan@gmail.com (K.K.V.); 3Department of Food Quality and Safety, Institute for Post-harvest and Food Sciences, The Volcani Center, Agricultural Research Organization, Rishon LeZion 7528809, Israel; mkswings321@gmail.com

**Keywords:** sugarcane, *Sporisorium scitamineum*, smut, proteomics, RT-qPCR, ISR

## Abstract

Smut disease is caused by *Sporisorium scitamineum*, an important sugarcane fungal pathogen causing an extensive loss in yield and sugar quality. The available literature suggests that there are two types of smut resistance mechanisms: external resistance by physical or chemical barriers and intrinsic internal resistance mechanisms operating at host–pathogen interaction at cellular and molecular levels. The nature of smut resistance mechanisms, however, remains largely unknown. The present study investigated the changes in proteome occurring in two sugarcane varieties with contrasting susceptibility to smut—F134 and NCo310—at whip development stage after *S. scitamineum* infection. Total proteins from pathogen inoculated and uninoculated (control) leaves were separated by two-dimensional gel electrophoresis (2D-PAGE). Protein identification was performed using BLASTp and tBLASTn against NCBI nonredundant protein databases and EST databases, respectively. A total of thirty proteins spots representing differentially expressed proteins (DEPs), 16 from F134 and 14 from NCo310, were identified and analyzed by MALDI-TOF/TOF MS. In F134, 4 DEPs were upregulated and nine were downregulated, while, nine were upregulated and three were downregulated in NCo310. The DEPs were associated with DNA binding, metabolic processes, defense, stress response, photorespiration, protein refolding, chloroplast, nucleus and plasma membrane. Finally, the expression of CAT, SOD, and PAL with recognized roles in *S. scitamineum* infection in both sugarcane verities were analyzed by real-time quantitative PCR (RT-qPCR) technique. Identification of genes critical for smut resistance in sugarcane will increase our knowledge of *S. scitamineum*-sugarcane interaction and help to develop molecular and conventional breeding strategies for variety improvement.

## 1. Introduction

Sugarcane (*Saccharum* spp.) is one of the most important industrial sources for crystal sugar, and it is cultivated across the world in tropical and subtropical countries. It is also the second largest biofuel crop and is an important source for many biomaterials. China is the fourth largest sugar producer in the world and the Guangxi province accounts for 92% of sugar production in China [1,2]. Sugarcane diseases are caused by bacteria, fungus, nematodes, virus, protozoa, phytoplasma, etc. with fungal diseases becoming a dominant group. Sugarcane smut disease, caused by the basidiomycete fungus *Sporisorium scitamineum* (Syn. *Ustilago scitaminea*), is a major sugarcane disease worldwide, and it can cause a 20–50% loss in cane yield [3], and up to 75% reduction in sugar production [4,5]. The fungus infects plants mainly through germinating buds in the soil or buds on standing stalks and grows in the plant in close association with the growing points or meristems, showing the presence of elongated whip, thin stalks, profuse tillering, and small narrow leaves.

Smut infection might also take place through the open stomata in leaves and the open areas in buds or wound in plant tissues and it is hard to control with chemicals in commercial crops [6]. To date, resistant varieties are the only practical, environmentally benign solution for managing sugarcane smut [7]. Breeding of smut-resistant sugarcane cultivars is a more economical and efficient approach to control the disease as compared to chemical treatments and agronomic practices [5,8,9]. Crop protection by modern genetic engineering technology is a potential tool to generate smut resistant varieties [10,11]. However, the complexity of sugarcane genetic background and the commercial viability of transgenic solutions for smut resistance make the genetic modification option unattractive [9,12]. Therefore, a better understanding of the biology, genetics, and molecular biology of smut resistance will greatly facilitate breeding sugarcane varieties for smut resistance.

Resistance mechanisms of sugarcane to smut involve external and internal disease resistance [13,14], and both mechanisms may confer resistance individually or in combination [15]. The external resistance is achieved by a physical barrier resulting from a mixture of bud structural characteristics [16], the thickness of the bud scales and chemicals such as phenyl-propanoids and glycosyl-flavonoids [17,18,19]. In the case of internal resistance, expressed after the pathogen attacks and penetrates through the bud scale: this is by several defense responses as well as increased lignin concentration [20], production of glycoprotein, phytoalexins, and polyamines [21,22,23,24]. At present, the exact nature of internal molecular defense mechanisms induced by smut remains less studied [15]. 

In previous reports, biochemical and genomics aspects of smut resistance were investigated, but not so much from a host–pathogen interaction perspective. A series of biochemical and molecular responses, such as triggering of specific defense signal transduction pathways, secretion of pathogenesis-related (PR) proteins and phytoalexins, and oxidative bursts occur in plants, during the stage of pathogen attack and the subsequent plant–pathogen interaction [25,26,27]. Next-generation sequencing based approaches were used to analyze the total changes in transcripts of resistant and susceptible sugarcane cultivars during sugarcane–*S. scitamineum* interaction [28,29]. Proteomic approaches offer powerful tools to study the expression of proteins and their function associated with plant-microbe interactions etc. [30]. One-dimensional gradient polyacrylamide gels (1DE), 2DE, and MS methods have been previously utilized to analyze the sugarcane proteome under various abiotic and biotic stresses [31]. Barnabas et al. [32] reported a total of 53 sugarcane differentially expressed proteins (DEPs) related to defense, stress, protein folding, and cell division. In addition, a putative effector of *S. scitamineum* pathogenesis, chorismate mutase, was found in sugarcane after *S. scitamineum* inoculation. The identification of DEGs during pathogen interaction is essential to enhance our understanding of plant resistance and offer clues as to what kind of defensive and biochemical mechanisms being regulated in a particular situation.

Therefore, the present study was mainly aimed at understanding the proteomic changes in the leaf of two sugarcane varieties with contrasting smut resistance (F134- resistant to *S. scitamineum* race 1 but susceptible to race 2 and NCo310- resistant to *S. scitamineum* race 2 but susceptible to race 1), which were planted after artificial inoculation with smut pathogen (susceptible to race 1 and race 2). The changes in contents of phyto-hormones (cytokinin (CYT) and ethylene (ETH)), as well as changes in the activity and expression of antioxidant enzymes (superoxide dismutase (SOD), catalase (CAT), and phenylalanine ammonia lyase (PAL)), were analyzed at different time intervals during the interaction. This also included a detailed study of proteome level alterations, gene expression, as well as biochemical changes sugarcane and smut pathogen interaction. To the best of our knowledge, the current study is the first comprehensive comparative proteomic analysis of sugarcane–smut pathogen interaction process in varieties with contrasting smut resistance after whip appearance stage. Expression of genes thought to be important for the pathogenesis is quantified to validate the proteomic data. The results obtained from this study clearly advance our molecular understanding of smut resistance in sugarcane, providing leads for identifying candidate genes and molecular markers for smut resistance.

## 2. Results

### 2.1. DE Analysis of Differentially Expressed Proteins in Sugarcane after S. scitamineum Inoculation

The infection of sugarcane plants inoculated with teliospores was first detected by a positive PCR reaction for molecular detection of *S. scitamineum* with pathogen species-specific primer in both varieties F134 and NCo310 (Appendix A), and it was further confirmed by anatomical changes observed between the healthy and infected plantlets under TEM. Figure 1 presents a 2D gel profile showing DEPs observed in control and smut-inoculated sugarcane plantlets, in which alteration was apparent for in the order of 80% of the spots. In the present study, a total of thirty DEPs were found, sixteen in F134 and fourteen in NCo310. Out of these proteins, the expression of four of them (spots 3, 4, 6, and 9) were upregulated and twelve (spots 1, 2, 7, 8, 16–19, and 21–24) were downregulated, as shown in Table 1. Whereas, in NCo310, the expression of eleven proteins (spots 1–4, 6, 7, 9, 10, 21, 23, and 24) were upregulated and three (spots 11, 20, and 22) were downregulated, as shown in Table 2. The identity of all the above proteins was established except for those located in spots 3, 8, and 21 in F134 and spots 3 and 21 in NCo310. All DEPs were individually collected for matrix assisted laser desorption ionization time-of-flight mass spectrometry (MALDI-TOF-TOF/MS) analysis.

### 2.2. MALDI-TOF-TOF/MS Analysis of Differentially Expressed Proteins

The results of MALDI-TOF-TOF/MS analysis of sixteen and fourteen differentially expressed proteins in F135 and NCo310 sugarcane varieties are shown in Appendix A. In both varieties, the function of some proteins was not identified. In infected F134, three spots (16, 17, and 19), and two spots (23 and 24) were identified as the same protein, whereas three spots (3, 8, and 21) were not identified (Table 1). However, in infected NCo310 variety two spots number (7 and 10), (23 and 24) were identified as the same protein, whereas two spots (3 and 21) were not identified (Table 2).

Peptide mass fingerprinting and tandem mass spectra of thirty proteins were achieved (Figure 2). Thirteen and twelve proteins had known functions and the sequence similarity was known to those proteins in both varieties. Based on bioinformatics analysis these proteins were found to be related to molecular processes, cellular components and categorized into numerous functional groups, i.e., peroxidase activity, DNA binding, metabolic processes, defense and stress responses, photorespiration, protein refolding, chloroplast thylakoid membrane, nucleus, plasma membrane, chloroplast, and proton-transporting ATP synthase complex (Appendix A).

### 2.3. Genes Expression Analysis by Real-Time Quantitative PCR (qRT-PCR)

In the present study, gene expression of CAT, SOD, and PAL were studied by qRT-PCR in leaf tissues at different stages of crop growth (Figure 3). The data showed a significant change in the expression level of all the three selected genes in both sugarcane varieties in response to smut infection. In comparison, the expression level of CAT increased significantly during the 180 days after planting in both F134 and NCo310. The maximum difference in CAT expression was observed between the two varieties at 30 days as compared to 60 and 90 days after planting. The difference in CAT expression between 60 and 90 days after planting in both varieties was not significant (Figure 3A). In response to smut infection, the expression patterns of SOD were similar in F134 and NCo310 varieties and continuously increased from day 30 to 180 sampling period after planting, although there was a dip in the expression on day 90 in both varieties (Figure 3B). There was no consistent expression pattern for PAL in both sugarcane varieties (Figure 3C). For F134, a significant increase in PAL expression was observed at 30 and 180 days, while it was decreased on day 60 and 90. There was no significant change in the level of PAL expression in NCo310 (Figure 3C). The comparative expression level of all the three genes was calculated and the results indicate that the highest expression of CAT, SOD, and PAL was approximately 20, 3, and 28 times at 180, 60, and 30 days, respectively, while the lowest (3, 2, and 3 times) was observed on day 90, 180, and 90 days in F134 as compared to NCo310.

### 2.4. Induction of Antioxidant Defense System by S. Scitamineum Infection

The activities of antioxidant enzymes SOD, PAL and CAT, and hormone contents were calculated for different time intervals (30–180 days) in both F134 and NCo310 following smut infection. Both leaf and root tissues were separately tested to measure the enzyme activities (Figure 4).

The data revealed that *S. scitamineum* inoculation caused an increase in the SOD activity in F134 sugarcane variety. In the leaf tissue of F134, the SOD activity was highest at 30 days (17.54 U.g^−1^FW) after planting and then it decreased with time. In NCo310, the activity of SOD was lower as compared to the control (Figure 4A). For root, the activity of SOD in F134 variety showed a small increase at 180 days, but in general, both varieties showed a reduction in SOD activity (Figure 4B). The SOD activity was higher in leaf, after the smut pathogen treatment for F134 than the control and NCo310.

After *S. scitamineum* infection, the initial activity of PAL in the leaf showed a significant increase in both varieties and that trend was observed up to 90 days as compared to control (Figure 4C). The PAL activity was significantly higher in root for both varieties and peaked at 90 and 180 days after infection in NCo310 and F134, respectively. (Figure 4D).

The CAT activity in the leaf did not significantly differ in the first 30 days following inoculation but it increased subsequently with the maximum activity occurring on day180 in F134 (34.26 U.g^−1^FW) and on day 60 in NCo310 (18.10 U.g^−1^FW) compared with the control (Figure 4E). In the root system of both sugarcane varieties, a significant difference in CAT activity was found compared to the control. In F134, the maximum CAT activity was observed on day 90 following smut. But in the case of NCo310, except at 60 days, an increasing trend in the CAT activity was noticed for the other time intervals (Figure 4F).

### 2.5. Phytohormone Levels as Affected by S. Scitamineum Infection

The levels of different hormones (ethylene and cytokinin) in roots and leaves of infected and control sugarcane plants showed a large, significant difference between treatments. The treatment of plants with *S. scitamineum* led to elevated cytokinin levels in leaves at different time intervals. In comparison with the control, the highest cytokinin activity was recorded at 90 and 180 days in F134, and it was increased with time reaching a maximum of 9.87 ng.g^−1^FW; whereas, in the infected NCo310, a lower level of cytokinin was produced as compared to control (Figure 5A). In the root tissue, change in cytokinin content was less and nonsignificant as compared to control except for that at 60 days in F134 and at 60 and 180 days in NCo310 (Figure 5B).

For ethylene, the level was lower in the leaves and higher in the roots as compared with the control (Figure 5C,D). The ethylene level was significantly higher in the leaf of F134 at 60 and 90 days than that in the control and showed a maximum (208.24 ng.g^−1^FW) at 90 days. In NCo310, the level of ethylene was not significantly different to that of control (Figure 5C). In the root, the level of ethylene was increased in both varieties as compared to the control at each time interval expect for that at 180 days in F134, A maximum hormone level was observed at 30 days (704.26 ng.g^−1^FW) followed by 90 days in F134 (Figure 5D). Whereas, in NCo310, the ethylene level showed a maximum value (284.30 ng.g^−1^FW) at 90 days.

## 3. Discussion

Proteomics based studies of plant–pathogen interaction contributes to a better understanding of the molecular and biochemical aspects of plant diseases. Plant–pathogen interaction is very complex due to the interaction of morphological, environmental, physiological, molecular, and metabolic factors with the pathogen. [31]. In the present research, for the most part, discusses the proteomic and gene expression responses occurring between two sugarcane varieties—F134 and NCo310—after the appearance of smut symptoms. A total of 16 DEPs in F134 and 14 in NCo310 were observed compared to their controls. After MALDI-TOF/TOF analyses these proteins were classified into different categories based on their association with various molecular, biological, and cellular processes.

Defense associated proteins: Heat shock protein 70 (HSp70), thioredoxin/transketolase fusion protein, putative thioredoxin peroxidase 1, and Cu/Zn superoxide dismutase are defense-related proteins that were differentially expressed and identified in this study. Upregulation of both HSp70 (spots 2 and 6) and thioredoxin/transketolase fusion protein (spot 4) was shown in NCo310. However, downregulation of HSp70 (spot 2), putative thioredoxin peroxidase 1 (spots 16, 17, and 19), and Cu/Zn superoxide dismutase (spot 22) was observed in F134, which suggest that in smut resistant sugarcane stress and defense-related proteins were upregulated during *S. scitamineum* infection as a defense strategy. HSP is a group of specific, conserved, and ubiquitous proteins distributed in the nucleus, cytoplasm, endoplasmic reticulum, mitochondria, and chloroplasts, and play an essential function in maintaining cellular functions when plants are subject to a variety of biotic and abiotic stresses such as heat stress, high salt, and heavy metal contamination [33,34]. In this research, the upregulation of two HSP70 proteins in NCo310 may be related to sugarcane defense response to smut invasion to protect the cellular structure, participate in denatured proteins refolding, maintain cell homeostasis and repair cellular dysfunction [35]. Previous studies also found that Hsp70s were upregulated under stress conditions during sugarcane–*S. scitamineum* interaction [6,36]. Downregulation of defense responsive proteins HSP-70 and DNAK-HSP 70 was observed in smut infected meristem cells at whip emergence stage by Barnabas et al. [32].

Peroxidases played important functions in defense and stress mechanisms and could be induced by several other physiological processes such as auxin catabolism, biosynthesis of lignin, cell wall stability, and senescence [37,38]. Many previous studies confirmed that reactive oxygen species (ROS), for example, hydrogen peroxide (H_2_O_2_), hydroxyl free radical (OH), and superoxide anion (O_2_^−^), are involved in the early resistance response in plants against pathogen attack [6]. In this study, the induction of three oxidative stress associated proteins—thioredoxin/transketolase fusion protein, putative thioredoxin peroxidase 1, and Cu/Zn superoxide dismutase—in F134 were observed. Likewise, the involvement and increased abundance of defense-related proteins—NTR, GST 1, STP, MDH, BQR, and SOD—accumulated probably because of damage related by means of extreme intra- and intercellular colonization, oxidative burst reaction of the host, and perturbance of normal cellular processes of sugarcane in infected meristem cells during whip emergence [32]. Thioredoxin-dependent peroxidases scavenge the excessive ROS and defend the sugarcane from smut pathogen attack [6].

Photosynthesis-associated proteins: Photosynthesis is an essential physiological process of the plant which plays a vital role in the development of the C_4_ crop. RuBisCO is a key enzyme in photosynthesis and a heteropolymer consisting of eight large subunits (RbCLs) and eight small subunits (RbCSs), which regulates photosynthesis and light respiration [39]. RuBisCO activity could be induced in several biotic and abiotic stress conditions [40,41] and increases photorespiration plus ROS production: the essential component of the hypersensitive defense response. The addition of these toxic components impairs cell death suppression and counteracts the efficiency of plant defenses to control pathogen infection [42]. In the present research, the expression of RuBisCO small subunit protein (spots 23 and 24) was upregulated in F134 and downregulated in NCo310, which suggested that upregulation of this enzyme may improve the growth of the NCo310 sugarcane variety with increased smut resistance. Similarly, two RuBisCO large subunits and one RuBisCO small subunit were upregulated after smut pathogen infection [6]. The expression of photosynthesis-related proteins was upregulated during the sugarcane and *S. scitamineum* interaction, which was favorable for the protection of the photosynthetic system in opposition to pathogen attack [36]. A *Nicotiana benthamiana* RuBisCO small subunit also played a vital role in tobacco virus movement and plant antiviral defense [43]. RuBisCO was upregulated significantly in *Rosa roxburghii* Tratt resistance to powdery mildew infection [44].

Pyruvate, orthophosphate (Pi) dikinase (PPDK) (spot 1), which is another photosynthesis-related protein, was also found to be upregulated in NCo310, whereas it was downregulated in F134 after *S. scitamineum* interaction. PPDK is a chloroplastic C_4_ cycle enzyme, catalyzes the ATP- and Pi-dependent formation of phosphoenolpyruvate (PEP), the primary CO_2_ acceptor molecule [45]. Chen et al. [46] observed that PPDK protein was considerably downregulated in maize responding to sugarcane mosaic virus (SCMV) infection using the first systemically infected leaves.

Other functional proteins: Translational elongation factor Tu (EF-Tu) is a protein that plays an essential function in the elongation phase of protein synthesis in plastids in plants. Spots 7 and 10 in NCo310 and spot 7 in F134 were identified as translational elongation factor Tu, and upregulation was observed in both. Fu et al. [47] indicated that EF-Tu plays a key role in the mechanisms of disease resistance and heat tolerance in plants. Nucleic acid binding protein 1 (spot 9) was upregulated in both varieties. In NCo310 variety, spot 20 identified as adenosine diphosphate glucose pyrophosphatase protein was downregulated but not expressed in F134. AGPPase catalyzes the hydrolytic breakdown of ADP glucose (ADPG) to produce equimolar amounts of glucose-1-phosphate and AMP in both mono- and dicotyledonous plants [48]. The induced expression of all the above proteins in sugarcane was useful for resisting the *S. scitamineum* infection.

Unknown and hypothetical proteins: The expression of two unknown proteins (spot 3 and 21) were also observed in both varieties, which may play a role in smut resistance in sugarcane. One hypothetical protein (spot 18) in F134 along with Os12g0277500 (spot 11) and Os01g0675100 (spot 22) were also identified in NCo310.

The RT-qPCR method was used to compare the expression of antioxidant enzymes (CAT, SOD, and PAL) at different developmental stages in both sugarcane varieties after smut pathogen interaction. The expression of all these enzymes was constantly elevated in F134 than NCo310, showing a positive response against disease resistance. In previous reports, PAL, which catalyzes an important step in the phenylpropanoid pathway, participated in sugarcane resistance to smut [32,49] and also played a role in resistance to chilling, drought, and salt stresses in sugarcane [50]. The activity of catalase, an iron porphyrin enzyme, was always higher in a smut resistant sugarcane variety than a susceptible variety, which protected sugarcane against reactive oxygen-related stimuli [51]. The expression of three different maize catalase genes was regulated differentially in response to fungal toxin [52]. Jain et al. [53] reported higher activity of SOD protects cells against ROS in water deficit conditions.

Phytohormones cytokinin and ethylene play an essential role in plants against the pathogen attacks [54]. In the present study, the different levels of both hormones were observed in leaves and roots, and their increased levels in F134 variety suggest their possible involvement in defense response. According to Rivero et al. [55], transgenic plants over producing cytokinins protected the plants from the harmful effects of abiotic stresses. Ethylene synthesis as a response to different stresses [56] is typically associated with various environmental stresses including in the resistance response of sugarcane to *S. scitamineum* [57,58].

In conclusion, the present study reports the proteomic responses of two sugarcane varieties with contrasting resistance to smut infection, F134 and NCo310, to *S. scitamineum* infection. The results showed significant DEPs expression in both varieties, and also in the plants inoculated with *S. scitamineum*. A total of 30 proteins including four upregulated and nine downregulated in F134, and nine upregulated and three downregulated in NCo310 after smut infection were identified. The protein peptide mass finger printing and tandem mass spectra of these proteins were successfully obtained in both verities. Bioinformatics investigation discovered that the functions of these 30 DEPs were related to various molecular and cellular functions associated with pathogenesis and plant defense mechanisms. The identified proteins were categorized into functional groups involved in peroxidase activity, DNA binding, metabolic processes, defense, stress responses, photorespiration, protein refolding, chloroplast thylakoid membrane, nucleus, plasma membrane, chloroplast, and proton-transporting ATP synthase complex. This is the first report of the proteomic exploration of the interactions between sugarcane interactions and *S. scitamineum*.

## 4. Materials and Methods

### 4.1. Plant Material, Source of Inoculum, and Inoculation

Two sugarcane varieties (F134 and NCo310) with contrasting *Sporisorium scitamineum* susceptibility were used in this study. F134 is resistant to *S. scitamineum* race 1 but susceptible to race 2 and NCo310 is resistant to *S. scitamineum* race 2 but susceptible to race 1. For the isolation of teliospores, mature plants of sugarcane variety ROC22 (susceptible to *S. scitamineum* race 1 and 2) were cut into 10 to 20 cm below the shoot top and placed in a sterile polythene bag. The plantlets of all sugarcane varieties were provided by Sugarcane Research Institute, Guangxi Academy of Agricultural Sciences, Nanning, China. The suspensions of teliospore were made by adding 0.1 g of *S. scitamineum* teliospores into 100 mL of sterile distilled water with a drop of Tween 20 and mixed properly with a magnetic stirrer [15]. The suspension of spore concentration was maintained to 5 × 10^6^ teliospores mL^−1^ by counting with a hemocytometer. The teliospores were incubated on potato dextrose agar (2%) at 30 ± 2 °C for 5 to 6 h to evaluate the germination rate. The viability of teliospores before inoculation was tested to confirm the sprouting ratio of *>* 90% [59]. The seedcanes of sugarcane varieties used in this study were cut into single-bud setts after removal of all the leaves and grown in the trays under controlled conditions (30 ± 2 °C, >80% relative humidity) for one month. For inoculation, one group of 30 healthy sugarcane plantlets was selected and immersed in *S. scitamineum* teliospores suspension for 2 h (treatment), and the other group of 30 sugarcane plantlets was treated with water as a control [1]. Then planted in pots (35 cm in diameter, 40 cm in depth) containing soil and sand mixture (3:1 *w*/*w*) in the greenhouse at Guangxi University, Nanning, China. Each variety had five sets of biological replicates with three plantlets in each replicate. Plants were arranged in a completely randomized design in the greenhouse. All sugarcane plants were irrigated once a day. Leaf and root samples were collected after 30, 60, 90, and 180 days of infection. Plant infection was confirmed by PCR based method and microscopic examination in both F134 and NCo310 [1]. All collected samples were immediately stored at −80 °C, until used for protein and RNA extraction.

### 4.2. Protein Extraction and Quantification

Sugarcane leaf samples were ground to fine powder under liquid nitrogen using a pestle and mortar. Extraction of total proteins followed the modified procedure described earlier [30]. Two grams of test sample powder were homogenized with 4 mL of cold extraction buffer (4 °C)([containing (g·L^−1^) Tris-HCl 0.25 M (pH-7.5), Sucrose 24%, EDTA-Na_2_ (Ethylene diamine tetra acetic acid disodium salt) 50 mM, SDS 2%, β-mercaptoethanol 2%, and PVP (Polyvinylpyrrolidone) 2%) and 8 mL of saturated phenol with Tris-HCl (0.25 M, pH 8.0) and β-mercaptoethanol (2%) was added before completing the maceration. Again 6 mL of extraction buffer was added and continued homogenizing till the preparation became a fine slurry. The homogenates were transferred to the centrifuged tube and mix properly for 1 min, and then centrifuged at room temperature at 14,000× *g* for 30 min. The supernatant was collected and re-extracted twice by adding an equal volume of extraction buffer without PVP in the centrifuged tube, followed by centrifugation at 14,000× *g* for 15 min. The supernatants were combined and proteins were precipitated overnight at −20 °C with cold methanol solution (1:5 *v*/*v*) (containing ammonium acetate 100 mM and β-mercaptoethanol 10 mM), then centrifuged at 12,000 × g for 10 min at 4 °C. The protein pellets were washed with cold methanol solution and centrifuged again at 4 °C to collect the protein pellets which were air dried under ice. The protein pellets were solubilized in rehydration solution (containing urea 8 M, CHAPS [3-(3-cholamidopropyl) dimethylammonio-1-propanesulfonate] 2% (*w*/*v*), thiourea 2 M, DTT (dithiothreithol) 40 mM, EDTA-Na_2_ 5 mM, and IPG buffer 1% (pH 4–7)) for 4 h at 28 °C. Finally, the proteins were centrifuged at 12,000 × g for 30 min at 4 °C to remove all undissolved particles and kept them at −80 °C. Total protein concentration was estimated according to the method described by Bradford [60] using bovine serum albumin (BSA) as the standard.

### 4.3. 2-DE, Image Acquisition, and Analysis

The immobilized pH gradient (IPG) gel strip (17 cm, pH 4–7, Bio-Rad) was carried out with first dimensional isoelectric focusing (IEF) on a PROTEAN IEF Cell apparatus (Bio-Rad). The protein extracts were dissolved at room temperature before rehydrating with rehydration solution containing urea 8 M, CHAPS 4%, DTT 65 mM, IPG buffer (pH 4–7) 0.2%, and Bromophenol blue 0.001%. The IPG gel strips were rehydrated at 18 ± 2 °C for 10 ± 12 h with 200 mL of rehydration solution and mixed with 500 μg protein. The protein was focused at 20 °C: 50 V for 12 h, 250 V for 30 min, 1000 V for 1 h, 3000 V for 2 h, 10,000 V for 2 h, to provide an overall of 60 kVh. When IEF was complete, the IPG strips were incubated for 15 min in the equilibrated buffer solutions containing DTT for reduction (Buffer I) and then the strips were re-equilibrated for 20 min in the equilibrated buffer containing iodoacetamide for alkylation (Buffer II) respectively, following Bio-Rad protocol.

The second-dimensional separation (sodium dodecyl sulfate polyacrylamide gel electrophoresis, SDS-PAGE) was implemented with the gel concentration of 12.5% T (Bio-Rad), and after solidification of the gel, the strips were placed and 0.5% of low melting agarose containing a drop of bromophenol blue was used for gel sealing. The gels were run at 40 V for 30 min and then 100 V until the bromophenol blue dye reached at the end of the gel on a Bio-Rad PROTEAN II system, and the plate temperature was maintained at 18 ± 20 °C by water flow from thermostatic circulator. When the SDS-PAGE was completed, the gel was kept out from the tank, stained with the BioSafe Coomassie (Bio-Rad), and destained with clean dH_2_O according to the manufacturer’s procedure. Stained gels were imaged by a Gel Doc 2000 (Bio-Rad) image scanner. Protein spots detection, spot matching, background subtraction, normalization, quantitative intensity, and statistical analysis were accomplished by using PD-Quest advanced 2D analysis software (version 8.0 Bio-Rad). The spots that exhibited as a minimum 2-fold-change were taken for the further experiment and differences were considered significant at *p* ≤ 0.05 level based on Student’s *t*-tests.

### 4.4. Mass Spectrometry and Data Analysis

The protein spots of interest on 2-DE gels were excised using Proteome Works Spot Cutter (Bio-Rad) with a 1.5 mm cutting diameter. After three times washing with Milli-Q water, peptide samples were prepared as by Song et al. [6]. The eluted samples were suspended in 0.1 % trifluoroacetic acid and spotted on a 384 well MALDI target plate through air drying until all solvent was evaporated. The peptides analysis was completed by 4700 MALDI TOF/TOF plus analyzer (Applied Biosystems Sciex, Foster, California, USA). The initial MS data was observed via reflector mode with the 4000-laser intensity. The MS spectra were collected in 2kV positive mode with fragments produced by collision induced dissociation. The scope of Peptide Mass Fingerprinting ranged from 800 to 4000 Da. GPS Explorer (Applied Biosystems Foster, California, USA) software was used for raw data search and Mascot as a search engine by NCBI (nr) protein database. Product mass tolerance was set at ± 0.3 Da with trypsin as a search parameter; alkylation and phosphorylation modifications were accepted. The match peptides with confidence intervals more than 98 were considered to be statistically significant, while peptides with lower scores were excluded. The function of identified proteins was determined by the Gene Ontology.

### 4.5. RNA Extraction, cDNA Synthesis, and qRT-PCR

To investigate the expression patterns of the associated genes, i.e., the gene for CAT, SOD, and PAL enzymes, at the time points of 30, 60, 90, and 180 days following *S. scitamineum* inoculation, tissue samples were collected from the two test varieties—F134 and NCo310—for RT-qPCR analysis. The glyceraldehyde-3-phosphate dehydrogenase (GAPDH) gene served as the reference gene. Total RNA was extracted from100 mg leaf tissue collected in triplicate from control and infected sugarcane varieties after symptom appearance with trizol reagent (Tiangen, Beijing, China) following the manufacturer’s guidelines. DNase I (Promega, Fitchburg, Wisconsin, USA) was used to eliminate the DNA impurity of RNA; the extracted RNA sample yield and purity were tested by a Nano photometer (Pearl, Implen-3780, Westlake, California, USA). One microgram of total RNA was used for single stranded cDNA synthesis by the Prime-ScriptTM RT Reagent Kit (TaKaRa, Dalian, China). Primers were designed as previously described with a reference gene [61] (Table 3). The specificity of primer sets was tested by melt curve examination and relative gene expression was determined by 2^−(Δ*C*t target gene − Δ*C*t reference gene)^ method [62]. The relative expression of both genes was calculated by the expression level of the infected sample minus the expression level of control at individual time intervals. Quantitative Real-Time PCR analysis was carried out in a Real-Time PCR Detection System (Bio-Rad, Hercules, California, USA) in SYBR Premix Ex Tap™ II (TaKaRa, Kyoto, Japan) with five replicates. Each 20 μL reaction mixture contained 2 μL template (10 x diluted cDNA), 10 μL SYBR Premix Ex Tap™ II, 0.8 μL of each primer (10 μM), and 6.4 μL ddH_2_O. For control, no RNA sample was used as the template. PCR conditions were 95 °C for 30 s, followed by 40 cycles of 95 °C for 5 s and 60 °C for 20 s in 96-well optical reaction plates. To confirm the specificity and amplification, a melting curve analysis was conducted. The relative quantification of *SuSOD*, *SuPAL*, and *SuCAT* genes to *GAPDH* was calculated by the 2^−ΔΔCt^ methods [62].

### 4.6. Determination of Biochemical Changes in Sugarcane

The quantitative changes in hormone (ethylene and cytokinin) content and enzyme (superoxide dismutase, phenylalanine ammonia lyase, and catalase) activity were estimated at 30, 60, 90, and 180 days after smut inoculation. Samples were randomly collected from both sugarcane varieties. Three replicates were used for all analyses. Fresh tissue samples were ground to make a fine powder under liquid nitrogen by using prechilled pestle and mortar. The measurements of hormone concentration level and enzyme activities were conducted by plant enzyme linked immune sorbent assays (ELISA) kit (Wuhan Colorful Gene Biological Technology Co., Ltd., Wuhan, China), following the manufacturer’s procedure [1].

#### 4.6.1. Antioxidant Enzyme Activities

Two grams of pulverized tissue samples from both treatment and control were homogenized in 9 mL of a 0.05 M phosphate buffer (pH 6.6) in a prechilled pestle and mortar. The homogenates were filtered through a C-18 extraction column and the filtrates were centrifuged at 15,000 for 20 min at 4 °C. The supernatant was collected and used for different enzymes activity analyses by plant ELISA kits. The whole extraction method was done at 4 °C. Briefly, ELISA was performed in 96-well microtiter plates coated with antigens against the selected enzymes. Forty microliters of test samples and 10 µL of antibodies were added in the antigen-coated wells and mixed gently. Along with, all three enzymes standard, blank and control wells was prepared separately according to manufacturer instructions, afterward, the plate was incubated for 30 min at 37 °C. The liquid in the plates was discarded after incubation, washed five times with washing buffer, and plates were air dried. Fifty microliters of HRP-Conjugate (different for each enzyme) reagent was added to each well of all plates, except blank well. Later, another time incubated for 30 min at 37 °C and washed as described above. After adding a chromogen solution, A and B (50 μL) to each well, kept in dark condition for color development for 15 min at 37 °C. The reaction was stopped by adding 50 μL stop solution to each well and color change was observed, i.e., from blue to yellow. The standard and control wells showed appropriate color development. For the assay of all enzymes, the plates were read on an ELISA Reader (Thermo Scientific, Multiskan GO, Waltham, Massachusetts, USA) at 450 nm within 15 min after the addition of stop solution. Enzyme activities were calculated with a standard curve and represented as U.g^−1^FW.

#### 4.6.2. Hormone Extraction and Assay

One gram each of powdered test samples (leaf and root) was mixed with 1 mL of 80% chilled methanol slowly in mortar and pestle and homogenized thoroughly and then kept at 4 °C for overnight. The mixture was then centrifuged at FF [12,000 rpm for 15 min at 4 °C. The collected supernatant was filtered through a C-18 extraction column. The extracted samples from the column were vaporized to remove the methanol under vacuum condition with ice, the samples were dissolved in phosphate buffer (pH 7.5), and the level of different hormone concentrations was measured by plant ELISA kit as described above.

### 4.7. Statistical Analysis

All biochemical activities were measured by the concentration level of treatment with the subtraction of their controls at each time interval. Standard errors were calculated for all mean values of three replicates and differences were considered significant at the *p* ≤ 0.05 level by Student’s *t*-test.

## Figures and Tables

**Figure 1 ijms-20-00569-f001:**
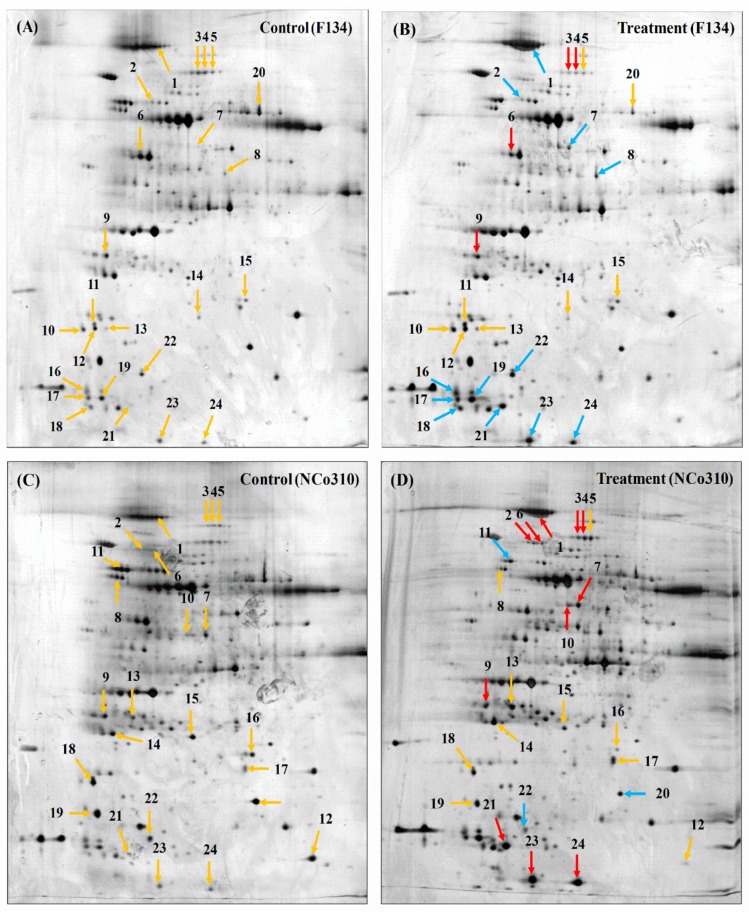
2-DE SDS-PAGE gel pictures of sugarcane varieties F134 and NCo310 with their controls. (**A**) control (F134), (**B**) treatment (F134), (**C**) control (NCo310), and (**D**) treatment (NCo310). Yellow for the protein spots of interest, red for upregulated proteins, and blue for downregulated proteins.

**Figure 2 ijms-20-00569-f002:**
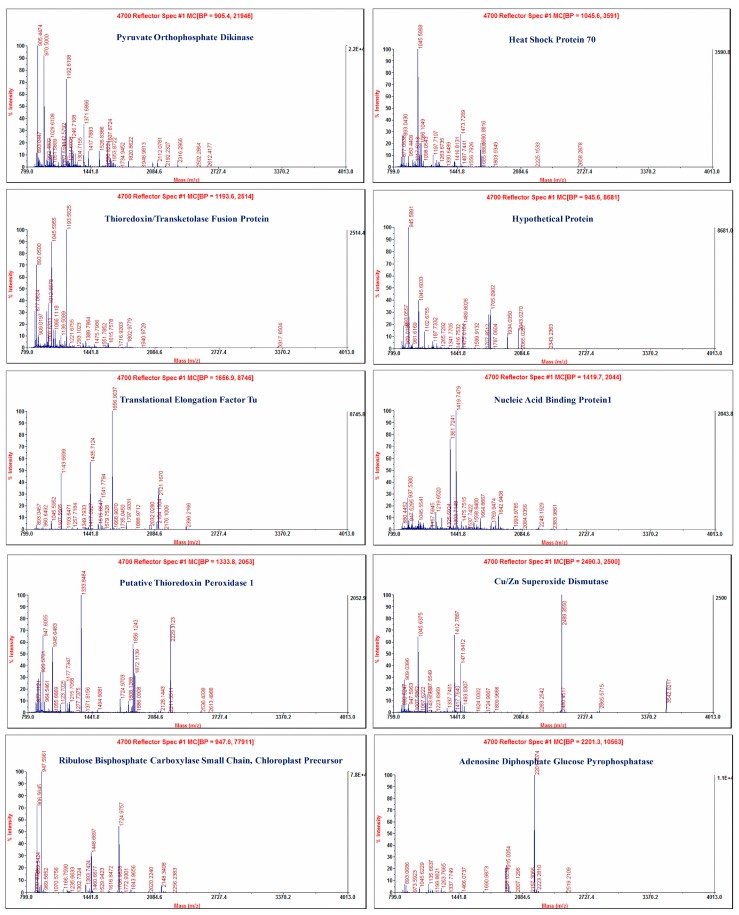
Peptide mass fingerprinting of differentially expressed proteins associated with sugarcane varieties F134 and NCo310 after *Sporisorium scitamineum* inoculation.

**Figure 3 ijms-20-00569-f003:**
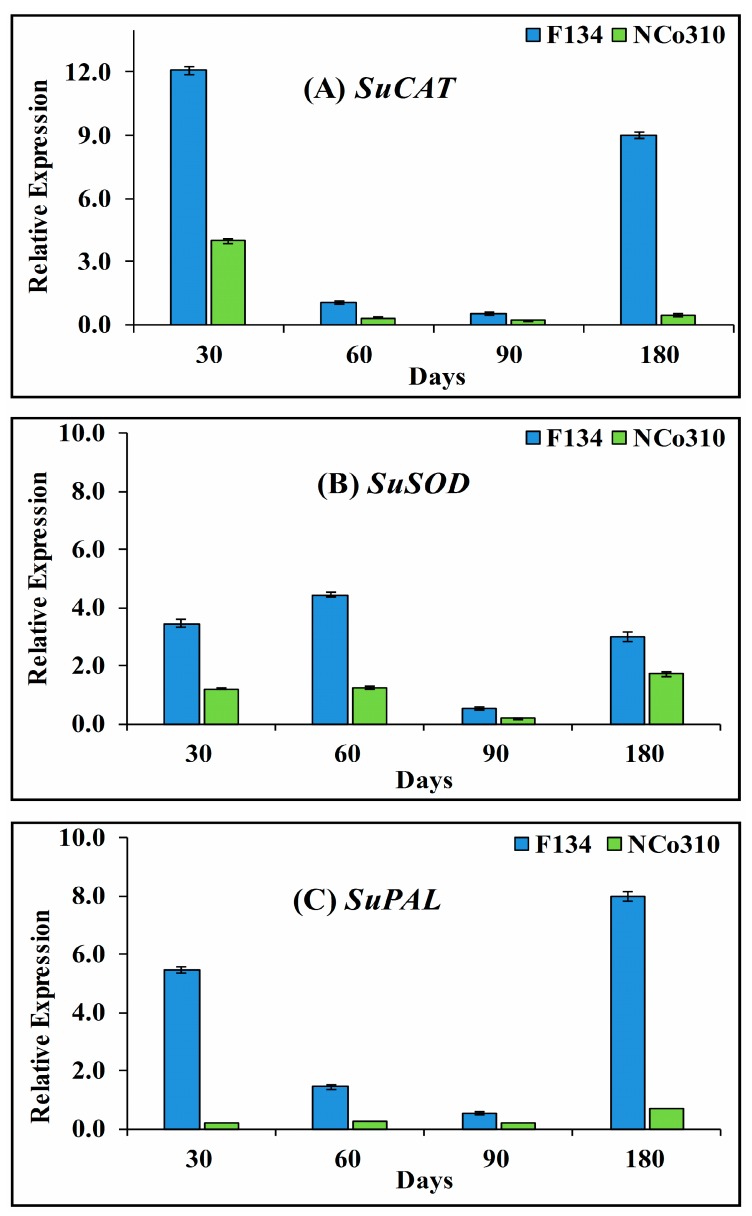
qRT-PCR analysis of differentially expressed genes in leaf tissue of sugarcane varieties F134 and NCo310 during sugarcane–*S. scitamineum* interaction. (**A**) Catalase (*SuCAT*), (**B**) superoxide dismutase (*SuSOD*), and (**C**) phenylalanine ammonia lyase (*SuPAL*). Data were normalized to the GAPDH expression level. All data points are the mean ± SE (*n* = 3).

**Figure 4 ijms-20-00569-f004:**
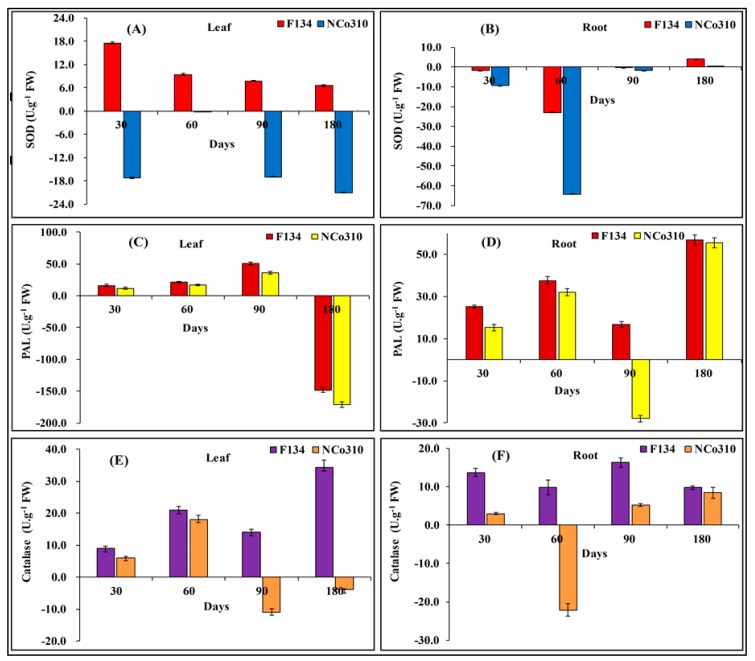
Analysis of enzyme activities in leaf and root tissues of sugarcane varieties F134 and NCo310 infected with *S. scitamineum* stress. (**A**,**B**) Superoxide dismutase, (**C**,**D**) phenylalanine ammonia lyase, and (**E**,**F**) Catalase. All data points (with the subtraction of their controls) are the mean ± SE (*n* = 3).

**Figure 5 ijms-20-00569-f005:**
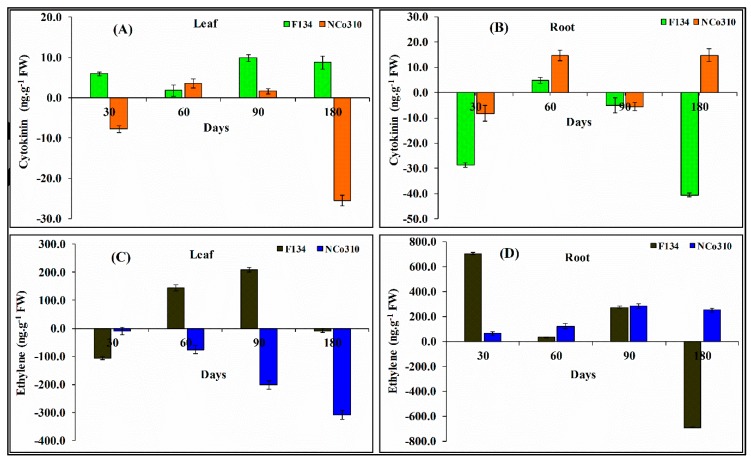
Analysis of phytohormone levels in leaf and root tissues of sugarcane varieties F134 and NCo310 infected with *S. scitamineum* stress. (**A**,**B**) Cytokinin and (**C**,**D**) ethylene. All data points (with the subtraction of their controls) are the mean ± SE (*n* = 3).

**Table 1 ijms-20-00569-t001:** Identification of differentially expressed proteins in variety F134 during the sugarcane–*Sporisorium scitamineum* interaction.

Spot No.	Accession No.	Identified Protein	Species	Protein				Expression
				Mw (Da)	(pI)	Score	Score C.I. (%)	
1	108796050	Pyruvate orthophosphate dikinase	*Saccharum officinarum*	102,293.9	5.5	977	100	Down
2	6911551	Heat shock protein 70	*Cucumis sativus*	71,444.1	5.07	111	99.994	Down
4	25067747	Thioredoxin/transketolase fusion protein	*synthetic construct*	86,860.6	5.59	173	100	UP
6	147843754	Hypothetical protein	*Vitis vinifera*	42,395.7	6.29	158	100	UP
7	17225494	Translational elongation factor Tu	*Oryza sativa*	50,381.8	6.19	477	100	Down
9	162463757	Nucleic acid binding protein1	*Zea mays*	33,096.6	4.6	87	98.709	UP
16, 17, 19	56182370	Putative thioredoxin peroxidase 1	*Saccharum officinarum*	10,780.7	4.9	317	100	Down
18	125543336	Hypothetical protein OsI_010708	*Oryza sativa* (indica cultivar group)	22,687.1	5.79	283	100	Down
22	1568639	Cu/Zn superoxide dismutase	*Triticum aestivum*	20,310.4	5.35	310	100	Down
23, 24	3914607	Ribulose bisphosphate carboxylase small chain, chloroplast precursor	*Saccharum officinarum*	19,023.5	9.04	464	100	Down

Spot numbers matched up to 2-DE gel in Figure 1; Expression relation was measured comparative to protein expression in control sample.

**Table 2 ijms-20-00569-t002:** Identification of differentially expressed proteins in variety NCo310 during sugarcane–*Sporisorium scitamineum* interaction.

Spot No.	Accession No.	Identified Protein	Species	Protein				Expression
				Mw (Da)	(pI)	Score	Score C.I. (%)	
1	108796050	Pyruvate orthophosphate dikinase	*Saccharum officinarum*	102,293.9	5.5	977	100	UP
2	6911551	Heat shock protein 70	*Cucumis sativus*	71,444.1	5.07	111	99.994	UP
4	25067747	Thioredoxin/transketolase fusion protein	*synthetic construct*	86,860.6	5.59	173	100	UP
6	56554972	Heat shock protein 70	*Medicago sativa*	70,952.1	5.08	144	100	UP
7, 10	17225494	Translational elongation factor Tu	*Oryza sativa*	50,381.8	6.19	334	100	UP
9	162463757	Nucleic acid binding protein1	*Zea mays*	33,096.6	4.6	87	98.709	UP
11	115488160	Os12g0277500	*Oryza sativa* (japonica cultivar-group)	61,092.6	5.12	316	100	Down
20	13160411	Adenosine diphosphate glucose pyrophosphatase	*Hordeum vulgare subsp. vulgare*	21,787.1	5.68	101	99.945	Down
22	115439131	Os01g0675100	*Oryza sativa* (japonica cultivar-group)	17,280.1	5.58	100	99.929	Down
23, 24	3914607	Ribulose bisphosphate carboxylase small chain, chloroplast precursor	*Saccharum officinarum*	19,023.5	9.04	464	100	UP

Spot numbers matched up to 2-DE gel in Figure 1; Expression relation was measured comparative to protein expression in control sample.

**Table 3 ijms-20-00569-t003:** Primers used in this study.

Gene	Primer	Sequence (5′-3′)	Strategy	Reference
*GAPDH*	GAPDH-F GAPDH-R	CTCTGCCCCAAGCAAAGATG TGTTGTGCAGCTAGCATTG	RT-qPCR	[61]
*SuPAL*	PAL-F PAL-R	CTCGAGGAGAACATCAAGAC GTGATGAGCTCCTTCTCG	RT-qPCR	[50]
*SuCAT*	CAT-F CAT-R	CTTGTCTGGAGCACATACACTTGGA TTCTCCGCATAGACCTTGAACTTTG	RT-qPCR	[63]
*SuSOD*	SOD-F SOD-R	TTTGTCCAAGAGGGAGATGG CTTCTCCAGCGGTGACATTT	RT-qPCR	[53]
*b*East mating-type	*b*E4-F *b*E8-R	CGCTCTGGTTCATCAACG TGCTGTCGATGGAAGGTGT	Genome PCR	[64]

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
