# Peer review of "Proteomic Analysis of the Resistance Mechanisms in Sugarcane during Sporisorium scitamineum Infection"

_ijms, 2019, doi:10.3390/ijms20030569_

Round 1

Reviewer 1 Report

The manuscript will merit publication only after a significative improvement of the quality of presentation.

In fact, the MS is extremely long, and some figures / data are not necessary.

Points to be corrected / modified:

-        Figure 1. Number are really too small; moreover, Arabic numbers could be in red for Up regulated proteins and blue for down regulated;

-        Figure 1 legend is not complete, Authors are invited to describe within treatment and indicate with A-D letters the four parts;

-        Table 2 (really Table 1) should be positioned following text line 141;

-        Table 2 and 3 (really Table 1 and 2): in the legend should be a copy of Figure 1 legend, whereas unknown proteins should be not included because the tables topic is the identification of the proteins Up or Down regulated that of course are only a part of the up of down regulated proteins;

-        Table 2 and 3 (really Table 1 and 2): “RuBisCO small subunit”, 4rt column, is not a Species, please indicate the real Species source;

-        Figure 2 and 3 should be moved in supplementary materials;

-        Line 162: it is not a validation but only an analysis of the SOD, CAT and PAL expression;

-        Figure 5: please use different colors for the histograms of the two cultivars; please add SuCAT, SuSOD and SuPAL in each frame; please modify “Day(s)” in “Days”.  

-        The legend of Figure 6 should be moved at Figure 7 and the other way around;

-        The Discussion is weak; also, it is not clear the connection between the proteins identified, and CAT/SOD/ PAL or phytohormones; please reduce the length of the Discussion, modify introducing a link between different results and add a physiological hypothesis to explain importance of the results obtained, otherwise the discussion appears as a mere list of data

Author Response

Response to Reviewer 1 Comments:

Point 1: The reviewers have suggested that your manuscript should undergo extensive English editing.

Response 1: The whole manuscript; language and grammar of the manuscript have been improved by a native English speaker and now it is as per the standards of the journal and changes made as suggested.

Point 2: In fact, the MS is extremely long, and some figures / data are not necessary.

Response 2: Changes made as suggested.

Point 3: Points to be corrected / modified:

Figure 1. Number are really too small; moreover, Arabic numbers could be in red for Up regulated proteins and blue for down regulated;

Response 3: Changes made as suggested.

Point 4: Figure 1 legend is not complete, Authors are invited to describe within treatment and indicate with A-D letters the four parts;

Response 4: Changes made as suggested.

Point 5: Table 2 (really Table 1) should be positioned following text line 141;

Response 5: Changes made as suggested.

Point 6: Table 2 and 3 (really Table 1 and 2): in the legend should be a copy of Figure 1 legend, whereas unknown proteins should be not included because the tables topic is the identification of the proteins Up or Down regulated that of course are only a part of the up of down regulated proteins;

Response 6: Changes made as suggested.

Point 7: Table 2 and 3 (really Table 1 and 2): “RuBisCO small subunit”, 4rt column, is not a Species, please indicate the real Species source;

Response 7: Changes made as suggested.

Point 8: Figure 2 and 3 should be moved in supplementary materials;

Response 8: Changes made as suggested.

Point 9: Line 162: it is not a validation but only an analysis of the SOD, CAT and PAL expression;

Response 9: Changes made as suggested.

Point 10: Figure 5: please use different colors for the histograms of the two cultivars; please add SuCAT, SuSOD and SuPAL in each frame;

Response 10: Changes made as suggested.

Point 11: please modify “Day(s)” in “Days”.

Response 11: Changes made as suggested.

Point 12: The legend of Figure 6 should be moved at Figure 7 and the other way around;

Response 12: Changes made as suggested.

Point 13: The Discussion is weak; also, it is not clear the connection between the proteins identified, and CAT/SOD/ PAL or phytohormones; please reduce the length of the Discussion, modify introducing a link between different results and add a physiological hypothesis to explain importance of the results obtained, otherwise the discussion appears as a mere list of data.

Response 13: The discussion part has been improved and changes made as suggested.

Reviewer 2 Report

The authors have used a traditional 2DE proteomics approach to measure the influence of fungal infection on different genetic strains of sugar cane.

In short the manuscript suffers from the following.

1)      No details of the MALDI approach

2)      No statistics of the differential changes

3)      No scores for protein detection

4)      Poor choose of phrases and words

5)      Great similarity to a 2011 paper, with insufficient clarification of what new knowledge is provided.

6)      Great similarity to a 2011 paper, in terms of the wording and layout.

7)      Failure to justify or integrate the PCR targets and phytohormone levels with proteomics data

Parts of the manuscript are almost well written but there seems to have been in many places an uninformed use of the thesaurus such that words have been replaced with thesaurus-like alternatives that are unsuitable. (Some examples are indicated below). Further, my concern if why would the authors resort to such alteration of regular terms. Notably the manuscript is very similar to reference 41. Moreover, in many places the structure of the manuscript has used reference 41 as a blue print for the layout, particularly in the introduction and discussion.

The two manuscripts are so alike the authors should make more emphasis on what additional information their work provides.

Also, there is no indication of what mass spectrometer was used or how the in gel digestion was performed.

For the protein identification there is no score, peptide numbers or sequence coverage provided for proteins identified. Nor is there any indication of the search algorithm used.

There are no statistics provided for the observed differences, i.e. fold changes and p-values. PDQuest  does, however, provide many measures.

Why is the time variation for the IPG strips so large? 10 +/- 12 h!

The use of “And” at the start of a sentence is not often recommended, particularly in scientific text, as it is here.

In many places long lists of terms are used followed by etc.  in a sloppy un-useful manner. For example where the associated ontology is listed.

The claim of genomic comparison only includes the selected targets (SOD, PAL & CAT). It was not clearly justified why these were chosen. At one point you claim that this measurement represents a validation, however, there is no validation of the differentially expressed proteins from the gels.

Examples of poorly chosen phrases or word choices

Page 1:  “Comprehensively, 30 DEP”

“ However, the nature of several applicable aspects still remain 20 unknown.”’

Page 2: “industry plays an essential responsibility”

“were mainly paid attention”

Page 7: “mass spectra were magnificently achieved,”

Page 15: “Appropriateness of 2DE coupled with MS, a reputable proteomic technique remains unchallenged till date especially in sugarcane as compared to other plants even though the arrival of numerous 252 strong high throughput proteomics tools”

“Understanding of plant-pathogen 255 interaction is very ‘composite’”

Page 16.” thioredoxin/transketolase fusion protein was observed in F134, which ‘recommended’ that in order..”

“superoxide anion (O2-) are ‘remarkably’ involved in the early resistance response”

“Que et al [41] also ‘recommended’ that”

Page 17:

“Earlier findings also ‘exposed’ that the activity of catalase”

“The ‘consequent’ protein peptide”

“This is the first repertoire of proteomic exploration information paying attention to the importance of the alterations of protein expression profile”

Page 19: “extracts were ‘permissible’ to melt at room temperature”

In appropriate use of tense:

Page 17: “It was believed that..”

Page 1: “ Available literature ‘suggested’ that..”

Finally, they validated the expression of DEPs”  (who are they?)

Page 2: “and assumed the existence of”

Typos;:

Page 2: “The external resistance is supposed to a physical wall”

Page 3: “The changes in contents of endogenous..”

Check the legend for Figure 6

Author Response

Response to Reviewer 2 Comments:

Point 1: The reviewers have suggested that your manuscript should undergo extensive English editing.

Response 1: The whole manuscript; language and grammar of the manuscript have been improved by a native English speaker and now it is as per the standards of the journal and changes made as suggested.

Point 2: The authors have used a traditional 2DE proteomics approach to measure the influence of fungal infection on different genetic strains of sugar cane.

In short the manuscript suffers from the following.

1) No details of the MALDI approach

2) No statistics of the differential changes

3) No scores for protein detection

4) Poor choose of phrases and words

5) Great similarity to a 2011 paper, with insufficient clarification of what new knowledge is provided.

6) Great similarity to a 2011 paper, in terms of the wording and layout.

7) Failure to justify or integrate the

The two manuscripts are so alike the authors should make more emphasis on what additional information their work provides.

Response 2:

1) Added in materials and methods section.

2) Added in materials and methods section.

3) Added in materials and methods section.

4) The whole manuscript has been checked and corrected by a native English speaker.

5,6,7) The present study is the first report to understand the changes in proteome occurring in two sugarcane varieties with contrasting smut resistance (F134- resistant to S. scitamineum race 1 but susceptible to race 2, and NCo310- resistant to S. scitamineum race 2 but susceptible to race 1) at whip development stage after S. scitamineum infection in Guangxi region, China. The identification of proteins critical for smut resistance in sugarcane will also increase our knowledge of S. scitamineum- sugarcane interaction and help to develop molecular and conventional breeding strategies for variety improvement.

Point 3: Also, there is no indication of what mass spectrometer was used or how the in gel digestion was performed.

Response 3: -Added in materials and methods section.

-The peptides analysis was completed by 4700 MALDI TOF/TOF plus analyzer (Applied Biosystems Sciex, USA).

-Gel digestion was performed by Song et al (2013) method.

Point 4: For the protein identification there is no score, peptide numbers or sequence coverage provided for proteins identified. Nor is there any indication of the search algorithm used.

Response 4: - Added in materials and methods section.

-The match peptides with confidence intervals more than 98 were considered to be statistically significant, while peptides with lower scores were excluded. The function of identified proteins was determined by the Gene Ontology.

-GPS Explorer (Applied Biosystems, USA) software was used for raw data search and Mascot as a search engine by NCBI (nr) protein database. Product mass tolerance was set at ±0.3 Da with trypsin as a search parameter, alkylation and phosphorylation modifications were accepted.

Point 5: There are no statistics provided for the observed differences, i.e. fold changes and p-values. PDQuest does, however, provide many measures. Why is the time variation for the IPG strips so large? 10 +/- 12 h?

The use of “And” at the start of a sentence is not often recommended, particularly in scientific text, as it is here.

Response 5: -Added in materials and methods section.

-The spots that exhibited as a minimum 2-fold-change were taken for the further experiment, and differences were considered significant at p ≤ 0.05 level based on Student’s t-tests.

-We optimized the 2D method in lab many times and we found the good results after rehydrated the IPG strips for 12 h which was already published by Yang et al (2011).

-Corrected

Point 6: In many places long lists of terms are used followed by etc. in a sloppy unuseful manner. For example where the associated ontology is listed.

Response 6: The whole manuscript; language and grammar of the manuscript have been improved by a native English speaker and now it is as per the standards of the journal and changes made as suggested.

Point 7: The claim of genomic comparison only includes the selected targets (SOD, PAL & CAT). It was not clearly justified why these were chosen. At one point you claim that this measurement represents a validation, however, there is no validation of the differentially expressed proteins from the gels.

Response 7: -In previous study, these genes were reported in stress response against smut pathogen, but not at different developmental stages in sugarcane. PAL, catalyzing an important step in the phenylpropanoid pathway, participated in sugarcane resistance to smut. The activity of catalase and SOD was higher in a smut resistant sugarcane variety than a susceptible variety, which protected sugarcane against reactive oxygen-related stimuli. Here, we first time report to compared the expression of these selected genes at different developmental stages in both sugarcane varieties after smut pathogen interaction.

-Yes, this is not a validation. We modified the sentence.

Point 8: Examples of poorly chosen phrases or word choices

Page 1: “Comprehensively, 30 DEP”

“However, the nature of several applicable aspects still remain 20 unknown.”’

Response 8:  The sentence has been modified.

Point 9: Page 2: “industry plays an essential responsibility” “were mainly paid attention”

Response 9: The sentence has been modified.

Point 10: Page 7: “mass spectra were magnificently achieved,”

Response 10: The sentence has been removed and modified

Point 11: Page 15: “Appropriateness of 2DE coupled with MS, a reputable proteomic technique remains unchallenged till date especially in sugarcane as compared to other plants even though the arrival of numerous 252 strong

high throughput proteomics tools”

“Understanding of plant-pathogen interaction is very ‘composite’”

Response 11: The sentence has been removed and modified

-The sentence has been modified.

Point 12: Page 16.” thioredoxin/transketolase fusion protein was observed in F134, which ‘recommended’ that in order.” “superoxide anion (O2-) are ‘remarkably’ involved in the early resistance response”

“Que et al [41] also ‘recommended’ that”

Response 12:  The sentence has been removed and modified

-The sentence has been modified.

Point 13: Page 17:

“Earlier findings also ‘exposed’ that the activity of catalase”

“The ‘consequent’ protein peptide”

“This is the first repertoire of proteomic exploration information paying attention to the importance of the alterations of protein expression profile”

Response 13: The sentences have been modified.

Point 14: Page 19: “extracts were ‘permissible’ to melt at room temperature”

In appropriate use of tense:

Response 14: The sentence has been modified.

Point 15: Page 17: “It was believed that.”

Response 15: The sentence has been removed and modified.

Point 16: Page 1: “Available literature ‘suggested’ that.”

“Finally, they validated the expression of DEPs” (who are they?)

Response 16: The sentences have been modified.

Point 17: Page 2: “and assumed the existence of”

Typos;

Page 2: “The external resistance is supposed to a physical wall”

Response 17: The sentences have been modified.

Point 18: Page 3: “The changes in contents of endogenous”

Response 18: The sentence has been modified.

Point 19: Check the legend for Figure 6

Response 19: Changes made as suggested.

Round 2

Reviewer 1 Report

Although I am not fully convinced of the relevance of the manuscript, the current version has certainly been correctly reviewed and the MS is now acceptable provided that some necessary changes are applied:

a)     Fig. 4 Legend; please modify in “ …. enzyme activities in of leaf and root tissues of sugarcane varieties …”;

b)     Fig. 4: activities must be expressed for FW not for ml;

c)     Fig.4 and Fig. 5: please use colours for histograms;

d)     450-455 M&M - Antioxidant Enzyme Activities: Authors must introduce the method applied for the enzyme assays and report them in literature. In fact, line 454, “… analyses by plant ELISA kits” is for enzymes completely wrong and misleading.

Author Response

Response to Reviewer 1 Comments:

Point 1: Fig. 4 Legend; please modify in “ enzyme activities in of leaf and root tissues of sugarcane varieties …”;

Response 1: Modified

Point 2: Fig. 4: activities must be expressed for FW not for ml;

Response 2: Changes made as suggested

Point 3: Fig.4 and Fig. 5: please use colours for histograms;

Response 3: Changes made as suggested

Point 4: 450-455 M&M - Antioxidant Enzyme Activities: Authors must introduce the method applied for the enzyme assays and report them in literature. In fact, line 454, “… analyses by plant ELISA kits” is for enzymes completely wrong and misleading.

Response 4: Added in the methodology section

Reviewer 2 Report

The revision appears adequate for publication. I have attached a marked up copy of the PDF with comments and errors indicated. These are mostly minor and in only a few places in the meaning unclear

Page 2 line 77, you talk about DEGs rather than DEPs

Line 63 and 89- rephrasing suggested

Page 3: line 100: and it was further confirmed

Line 103-4: Do you mean:
"in which alteration was apparent for in the order of 80% of the spots."

The section on the MALDI results is landscape and harder to read

Page 5: You state:"the same with unknown identity",  what do you mean? Please rephrase.

"(3 and 21) with unknown identity"- were these not identified or are they hypothetical proteins?

Page 6: What do you mean by: "The repeated protein peak value was the same for five proteins in F134, and for four proteins in NCo310."

Page 16:  20000 voltage? Is this 20KV or 2 Kv?

Page 17: What kind of statistical tests were used?

As a final comment should you be concerned that the heat shock proteins are only a general stress response and there mights be more important changes

Author Response

Response to Reviewer 2 Comments:

Point 1: Page 2 line 77, you talk about DEGs rather than DEPs

Response 1: Corrected

Point 2: Line 63 and 89- rephrasing suggested

Response 2: Line 63 and 89 was modified

Point 3: Page 3: line 100: and it was further confirmed

Response 3: Changed

Point 4: Line 103-4: Do you mean: "in which alteration was apparent for in the order of 80% of the spots."

Response 4: Yes, the sentence was modified

Point 5: The section on the MALDI results is landscape and harder to read

Response 5: Changes made as suggested

Point 6: Page 5: You state: "the same with unknown identity", what do you mean? Please rephrase.

Response 6: Sentence was modified as a function of some proteins was not identified

Point 7: "(3 and 21) with unknown identity"- were these not identified or are they hypothetical proteins?

Response 7: These were not identified, the sentence was modified

Point 8: Page 6: What do you mean by: "The repeated protein peak value was the same for five proteins in F134, and for four proteins in NCo310."

Response 8:  Sentence was modified

Point 9: Page 16:  20000 voltage? Is this 20KV or 2 Kv?

Response 8:  2 Kv, modified

Point 10: Page 17: What kind of statistical tests were used?

Response 10: Student t-test, added

Point 11: As a final comment should you be concerned that the heat shock proteins are only a general stress response and there mights be more important changes

Response 11: Yes, Heat shock proteins (HSP) are produced by cells generally in response to exposure to stressful conditions.